# Molecular Mechanisms Underlying Remodeling of Ductus Arteriosus: Looking beyond the Prostaglandin Pathway

**DOI:** 10.3390/ijms22063238

**Published:** 2021-03-22

**Authors:** Ho-Wei Hsu, Ting-Yi Lin, Yi-Ching Liu, Jwu-Lai Yeh, Jong-Hau Hsu

**Affiliations:** 1School of Medicine, Kaohsiung Medical University, Kaohsiung 80708, Taiwan; kennyhsu0426@gmail.com (H.-W.H.); lintingyi2014@gmail.com (T.-Y.L.); 2Department of Pediatrics, Kaohsiung Medical University Hospital, Kaohsiung Medical University, Kaohsiung 80708, Taiwan; a740624@ms42.hinet.net; 3Graduate Institute of Medicine, College of Medicine, Kaohsiung Medical University, Kaohsiung 80708, Taiwan; jwulai@kmu.edu.tw; 4Department of Pharmacology, College of Medicine, Kaohsiung Medical University, Kaohsiung 80708, Taiwan; 5Department of Medical Research, Kaohsiung Medical University, Kaohsiung 80708, Taiwan; 6Department of Marine Biotechnology and Resources, National Sun Yat-Sen University, Kaohsiung 80424, Taiwan; 7Department of Pediatrics, Faculty of Medicine, College of Medicine, Kaohsiung Medical University, Kaohsiung 80708, Taiwan

**Keywords:** patent ductus arteriosus, prostaglandin, vascular remodeling

## Abstract

The ductus arteriosus (DA) is a physiologic vessel crucial for fetal circulation. As a major regulating factor, the prostaglandin pathway has long been the target for DA patency maintenance or closure. However, the adverse effect of prostaglandins and their inhibitors has been a major unsolved clinical problem. Furthermore, a significant portion of patients with patent DA fail to respond to cyclooxygenase inhibitors that target the prostaglandin pathway. These unresponsive medical patients ultimately require surgical intervention and highlight the importance of exploring pathways independent from this well-recognized prostaglandin pathway. The clinical limitations of prostaglandin-targeting therapeutics prompted us to investigate molecules beyond the prostaglandin pathway. Thus, this article introduces molecules independent from the prostaglandin pathway based on their correlating mechanisms contributing to vascular remodeling. These molecules may serve as potential targets for future DA patency clinical management.

## 1. Introduction

During fetal development, the ductus arteriosus (DA) bridges the systemic circulation with the pulmonary circulation. DA is a physiologic vessel until birth. After birth, the neonate’s oxygenation and carbon dioxide elimination no longer rely on the maternal circulation, and the heart serves as the major bridge of systemic circulation and pulmonary circulation. Closure of the DA occurs spontaneously after birth in most infants. Nevertheless, a significant number of neonates fail to undergo DA closure. The prevalence of patent ductus arteriosus (PDA) was 7.31 per 10,000 births in Taiwan from 2005 to 2014 [1]. Another report showed an incidence of 1 per 2000 births in term neonates, but higher incidence rates, varying from 20% to 60%, were observed in preterm neonates [2]. 

The DA’s closure can be categorized into two phases: the initial functional closure and the subsequent anatomical closure [3]. The functional closure involves smooth muscle cell (SMC) constriction in response to increased arterial oxygen, the decline in prostaglandin, and decreased blood pressure [4,5]. In most term newborns, functional closure occurs in the first few hours after birth. The anatomical closure process requires a longer duration, demanding several weeks to complete. 

The prostaglandin pathway plays a major role in both the functional and the anatomical closure and is recognized as a valuable pharmacological target for DA patency maintenance [4,6]. Conversely, prostaglandin pathway inhibition through cyclooxygenase (COX) inhibitors is regarded as the mainstay therapy for PDA closure in premature infants [7]. Whether the patient’s PDA should be closed or maintained varies depending on different clinical conditions [8]. In selected cases with congenital heart diseases, DA patency provides sufficient oxygenation in the systemic circulation. On the other hand, untreated PDAs develop into heart failure and result in several complications such as pulmonary hypertension, arrhythmia, aneurysm, and infectious endocarditis [9].

The drawbacks of both prostaglandin and COX inhibitors limit clinical efficiency. Physicians struggle with prostaglandin’s adverse effects, such as respiratory depression, cutaneous vasodilation, temperature elevation, and seizure-like activities [10]. The most alarming side effect, respiratory depression, is most frequently observed in infants with lower birth weights and risks prostaglandin therapy’s discontinuance. Similarly, adverse effects of COX inhibitors indomethacin and ibuprofen include risks of oliguria, serum creatinine level elevation, and gastrointestinal bleeding [11,12]. Recently, acetaminophen had been proposed as an alternative to COX inhibitors for its relatively less adverse effects [13,14]. Further trials to investigate the efficacy and safety of acetaminophen are warranted.

Despite the repeated course of COX inhibitors, a proportion of patients failed to respond and underwent surgical intervention [15,16]. Our previous study had shown that an estimate of 25% of patients failed to respond to COX inhibitor therapy [17]. Several factors were associated with response failures, such as age, gender, application of surfactants, and genetic polymorphisms [18,19,20]. Aside from alterations in drug metabolism, prolonged use of COX inhibitors paradoxically inhibits the DA’s vascular remodeling, which may partly explain the response failure [21]. Hence, we intend to investigate beyond the prostaglandin pathways and highlight molecules that influence vascular remodeling to propose possible targets for future PDA management. 

## 2. Mechanisms of Anatomical Closure

Associated with vascular remodeling, the anatomical closure process involves five complex mechanisms, including (1) SMC migration and proliferation, (2) extracellular matrix (ECM) production, (3) endothelial cell (EC) proliferation, (4) internal elastic laminae (IEL) disruption, and (5) blood cell-related mechanisms. These mechanisms are sophisticatedly orchestrated and mutually intertwined. Total DA occlusion occurs following the completion of the two phases.

This article will focus on molecules affecting the DA’s vascular remodeling according to the aforementioned mechanisms in anatomical closure. These molecules are organized in Table 1.

### 2.1. SMC Migration and Proliferation

SMC migration and proliferation are important mechanisms in early intimal cushion formation, as shown in Figure 1. Prostaglandin stimulates SMC migration and hyaluronic acid deposition, amplifying the prostaglandin pathway [39]. This self-reinforcing pathway is crucial in promoting SMC migration into the subendothelial region [40]. Nonetheless, molecules independent from such pathways were also identified.

#### 2.1.1. SMC Migration

##### Notch

Notch is associated with the stimulation of SMC migration and proliferation. Its signaling pathway has a wide variety of developments and disease associations, from tissue growth promotion to tumor cell suppression [41]. These developments include the development of somite-derived organs, vessels, the heart, the hematopoietic system, the nervous system, the cochlea, and the pancreas [42]. Regarding vasculature, Notch signaling plays a role in endothelial cell differentiation, arteriovenous specification, vasculature branching, and the differentiation of vascular smooth muscle cells (VSMC). Notch signaling also regulates the homeostasis of VSMC. Whether Notch signaling promotes the transformation of VSMC into the contractile state or the synthetic state is still under debate [43]. Notch signaling pathway dysregulation is observed in various vascular diseases, such as pulmonary hypertension and retinal vascular disorder. Its role in ischemic stroke and dementia with hereditary characteristics was identified [44]. Furthermore, by regulating transforming growth factor-beta (TGF-β), Notch may contribute to the conversion from VSMC to chondrocytes, causing calcification of vessels [45]. Multi-organ diseases also arise from Notch mutations; for example, the Alagille syndrome, involving anomalies of various organs, correlated with inherited genetic mutations, and the Hajdu–Cheney syndrome, a rare genetic disorder that induces bone tissue malformation and causes several other organ abnormalities [46]. 

The Notch receptor is located at the cell membrane, with two domains—extracellular and intracellular domains (ICDs). When the receptor is activated, two stages of cleavage occur. The first proteolytic cleavage of the receptor is performed by the “a disintegrin and metalloproteinases (ADAM)” enzyme, followed by the second stage of cleavage conducted by the γ-secretase complex [47]. This results in the nuclear localization of the cleaved ICDs. After sophisticated interaction with intranuclear proteins, the ultimate complex formed may bind to certain sites of Notch target genes, attract coactivators, and therefore regulate gene expression; for instance, downstream HES1/2/5 genes (Figure 2) are activated in this manner, which correlates with SMC migration and proliferation in the DA.

The Notch signaling plays a role in PDA. Jagged1, a ligand that reacts with the Notch receptor, regulates integrin avb3 expression that affects VSMC adhesion to vessels [48]. The deletion of such protein-encoding gene was associated with PDA in mice [49]. Knockout of Notch2 and Notch3 also led to PDA in mice [43]. N-[N-(3, 5-difluorophenacetyl-L-alanyl)]-S-phenylglycine t-butyl ester (DAPT), a γ-secretase inhibitor, indirectly inhibits the Notch pathway and negatively affects SMCs’ proliferation and migration. Wu et al. noted the anti-oxidative effects of DAPT on DA SMCs in mice. More specifically, the inhibition of the Notch pathway inhibits angiotensin II-induced DA SMC migration and proliferation, correlated with decreased reactive oxygen species production, lower calcium influx, and attenuated ERK1/2 JNK signaling [22]. Li et al. also disclosed that the Notch downstream pathway of lower cytosolic calcium levels inhibits DA SMC proliferation and migration [50]. In conclusion, inhibition of the Notch signaling pathway inhibits SMC migration and proliferation by downregulating angiotensin II-related signaling.

##### Fibronectin

Fibronectin mediates SMC migration. A type of glycoprotein, fibronectin, is a ubiquitous ECM found in various cell types [51]. Fibronectin plays a role in cellular processes, hemostasis, and thrombosis [52]. Following arterial injuries, SMCs are activated, and fibronectin is deposited around proliferative cells in the media and intima: a phenomenon similar to neointimal formation [53].

There are two isoforms of fibronectin, the plasma fibronectin, and the cellular fibronectin. The cellular fibronectin is the predominant form, which is the form of fibronectin found in the ECM. After alternative splicing of the cellular fibronectin, one of the products is Fn-EDA, fibronectin containing extra domain A (EDA). Fn-EDA is secreted by endothelial cells and SMCs and acts as a ligand for integrins and Toll-like receptor 4 (TLR4) [54]. The signaling pathways of integrin and TLR4 are implicated in SMC migration and proliferation, suggesting that Fn-EDA may be involved in the modulation of neointima hyperplasia.

Mason et al. conducted in utero studies of fetal lamb DA. They discovered that augmented mRNA translation in SMCs increases fibronectin synthesis during intimal cushion formation, facilitating SMC migration into the subendothelium [23]. Interestingly, another study revealed findings of a high fibronectin concentration observed in PDA patients [24]. Apart from cellular fibronectin, plasma fibronectin may also function in DA closure. Previous studies have demonstrated that plasma fibronectin may display diverse platelet aggregation roles: stimulatory when fibrin is present, and inhibitory when fibrin is absent [52]. Low concentrations of fibrinogen were presented in these cases, and fibronectin is believed to exhibit inhibitory effects on platelet aggregation and thrombogenesis in these settings.

##### The Role of Retinoic Acid in Smooth Muscle Cell Migration and Proliferation

Retinoic acid (RA) can stimulate SMC migration and proliferation in the DA. Reviewing its function in development, the RA signaling pathway plays a role in developing several organs and tissues, such as the body axis, limbs, spinal cord, eye, heart, vasculature, and the reproductive tract [55,56]. As an active metabolite of vitamin A, there are several isoforms of RA, including all-trans retinoic acid (atRA), 9-cis-RA, and 13-cis-RA. It had been discovered that atRA functions as the primary ligand during development [57]. The signaling mechanism of RA had also been revealed. RA binds to the intra-nuclear RA receptor (RAR) and forms a heterodimer with the retinoid X receptor (RXR). These RAR-RXR heterodimers bind to RA-responsive elements, regulating the transcription of RA-targeted genes, activating or repressing.

In contrast to the stimulatory effects on the DA, atRA was found to suppress smooth muscle cell migration and proliferation in other vessels, including femoral arteries and carotid arteries. Zhang et al. disclosed the inhibitory effects of atRA on neointima hyperplasia through AMP-activated protein kinase in mouse carotid arteries [58]. Wiegman et al. also showed that atRA administration leads to less restenosis in rabbits with an atherosclerotic femoral artery that underwent balloon angioplasty [59]. Similarly, Herdeg et al. came to similar conclusions, noting that local atRA therapy limited restenosis in atherosclerotic rabbit carotid arteries after balloon angioplasty [60]. The distinct response to RA in the DA may be explained by the presence of a unique phenotype of SMC myosin heavy chain, SM2, in the DA SMCs. Colbert et al. discovered the precocious expression of SM2 in the DA, while SM1 was found in all embryonic vasculatures [44].

Retinoic acid stimulates platelet-derived growth factor (PDGF)-mediated SMC migration in the DA. Yokoyama et al. found that atRA alone did not significantly induce SMC migration. Nevertheless, when PDGF-BB is present, a significant increase in SMC migration after administration of atRA occurs [25]. Another study conducted by Goyal et al. showed similar results. They found that the RA receptor activation signal pathway expression was altered significantly in the DA, but not in the aorta [61]. Additional to migration, RA also stimulates SMC proliferation. Wu et al. discovered that RA stimulated the DA’s vascular SMCs’ growth but not that of the SMCs of the aorta or the pulmonary artery [26]. With these benefits above on DA closure, we should still bear in mind that though adequate RA may help prevent PDA, excessive exposure can cause SMCs to undergo apoptosis and lead to the cardiovascular system’s teratogenesis [26].

##### The Role of Transforming Growth Factor-Beta 1 in Smooth Muscle Cell Migration

On the other hand, transforming growth factor-beta 1 (TGF-β1) inhibits SMC migration. TGF-β1 is a vital cytokine in the vascular system’s development and physiology [62]. Belonging to the TGF-β family, it regulates cellular activities such as proliferation, migration, differentiation, and apoptosis [63]. Furthermore, TGF-β1 is also crucial in the inflammatory response. Recent studies have indicated its diverse cancer function, where it is tumor-suppressive in early stages and healthy cells, but tumor-promotive in later stages [64].

There are three forms of TGF-β, TGF-β1, TGF-β2, and TGF-β3, acting as ligands, and three forms of TGF-β receptors (TGFBR) TGFBR1, TGFBR2, and TGFBR3. The signaling pathways of TGF-β can be categorized into two pathways: the canonical pathway and the noncanonical pathway [65]. The canonical pathway involves the phosphorylation of TGFBR1, which in turn phosphorylates “mothers against decapentaplegic homolog (SMAD)”. SMAD then translocates to the nucleus and modulates gene transcription. The noncanonical pathway can activate numerous kinases and give rise to several downstream pathways. Both pathways can cross-talk with other signaling pathways, generating diverse cellular signals [63].

Tannenbaum et al. investigated the effect of TGF-β1 on DA SMCs of fetal lamb. In the DA SMCs, they respond to TGFβ1 by increasing a5b1 integrin on the cell surface, promoting SMC migration. Nonetheless, this effect is overridden by another effect, where integrin increases and anchors the cytoskeleton. The combinational result of the two converse effects is the inhibition of SMC migration [27]. This result differs from that of aortic SMCs; in vitro studies showed the stimulatory effect on migration [66].

#### 2.1.2. SMC Proliferation

##### The Role of Interleukin-15 in Smooth Muscle Cell Proliferation

Interleukin-15 (IL-15), a cytokine better known for its role in immune responses, has an anti-proliferative effect on SMC proliferation [67]. It is a pro-inflammatory cytokine critical for the development, survival, proliferation, and activation of several lymphocyte lineages [68]. Moreover, studies have also revealed its role in cancer (mostly anti-tumor) and autoimmunity [69,70]. Finally, a recent study revealed the association of inflammatory markers with PDA, including IL-15 [36].

Iwasaki et al. noticed that IL-15 lowered the expression of CX3CR1 mRNA in the SMCs of the DA [28]. They conducted experiments using rat DA SMCs under primary culture and examined the expression of CX3CR1 mRNA, comparing those exposed to IL-15 and those that were not. Since CX3CR1 assists in SMC proliferation, decreased expression of such signaling contributes to an inhibitory effect. Altogether, IL-15 exhibits an inhibitory effect on DA SMC proliferation through the downregulation of CX3CR1. 

##### PRDM6

PRDM6, a histone modifier gene, is crucial for SMC proliferation. It belongs to the PRDM family, a group of gene-encoding proteins that consist of a PR domain and zinc finger repeats. Research has revealed its role in vascular development. In specific vascular precursor cells, PRDM6 may alternate cell fate, promote smooth muscle cell differentiation, advocate endothelial cell apoptosis, and inhibit endothelial cell proliferation [71,72]. Furthermore, PRDM6 mutation significantly affected several goats’ growth traits, implying its growth and development role. 

PRDM6 is expressed in the SMCs and is vital for maintaining the proliferative state of the DA SMCs. Li et al. studied the effect of PRDM6 mutation in mice DA [29]. In the SMCs of mice DA with PRDM6 loss, early differentiation was observed, restricting proliferation. Increased arsenic exposure correlates with the suppression of PRDM6 expression [73]. This might explain the report showing significantly increased PDA in infants whose mother was exposed to arsenic in drinking water [74]. 

### 2.2. ECM Production

The ECs deviate from the internal elastic lamina during anatomical closure, creating a subendothelial region [75]. The ECM is produced and deposited in this region. Deposition of the ECM occurs in the normal closing DA but is absent in PDA. The ECM may assist in DA closure; for example, when hyaluronic acid accumulates, it promotes SMC migration into endothelial layers and, in turn, neointimal cushion formation [21]. 

#### 2.2.1. The Role of RA in Extracellular Matrix Production

RA had been found to increase ECM production in various conditions. In wound healing, RA stimulated the production of ECM, such as collagen type 1 and fibronectin [76]. Vitamin A deficiency is associated with impaired lung development via the modification of the ECM configuration [77].

RA increases fibronectin and hyaluronic acid (HA) production during DA closure. Yokoyama et al. brought to light that maternal vitamin A administration made up for the elevated number of transcripts of fibronectin 1 and hyaluronic acid synthase 2 (HAS2) in preterms. Such elevation confers to SMC migration and, likewise, intimal cushion formation [25]. Similar findings were demonstrated in the study conducted by Shanker et al., where RA increased fibronectin expression [78]. Saavalainen et al. also discovered that RA leads to increased HAS2 expression [79]. In conclusion, RA provided stimulatory effects on DA closure by stimulating SMC migration and proliferation and increasing ECM production.

#### 2.2.2. The Role of TGF-β in Extracellular Matrix Production

It has been well established that TGF-β mediates tissue fibrosis, the condition in which the overt or dysregulated ECM is deposited and disrupts the primary tissue structure and function [80,81]. Further, as previously mentioned, ECM-associated TGF-β plays a complex role in tumorigenesis [82,83]. 

TGF-β increases HA and chondroitin sulfate production in the DA [84]. Boudreau et al. discovered that TGF-β in ECs could promote glycosaminoglycan production (such as HA and chondroitin sulfate), but not fibronectin synthesis in DA SMCs. Yokoyama et al. also noted that TGF-β increases HA production in DA SMCs, though the extent is much less than prostaglandin pathways [21]. Interestingly, TGF-β1 exhibits an inhibitory effect on SMC migration as described above, implying that TGF-β may have adverse roles in vascular remodeling of the DA.

#### 2.2.3. The Role of PDGF-BB in Extracellular Matrix Production

PDGF-BB belongs to the PDGF family, where four distinct polypeptide chains give rise to five dimeric isoforms of PDGF, PDGF-AA, PDGF-AB, PDGF-BB, PDGF-CC, and PDGF-DD. These ligands and the two types of tyrosine kinase receptors, PDGF receptors α and β, form the PDGF network. PDGF functions as mitogens and also promotes angiogenesis, crucial for early embryonic development [85]. PDGF-BB secretion following injuries accelerates the healing process, but this requires further investigation [86]. 

Interestingly, PDGF-BB slightly increases HA production in DA SMCs. Regulation of PDGF-BB and its receptor has been known to play a role in pulmonary hypertension’s pathogenesis [87]. Utilizing the primary culture of rat DA SMCs, Yokoyama et al. unearthed that PDGF-BB enhances the expression of HA-synthesizing mRNA and consequently increases HA synthesis [21]. Papakonstantinou et al. also revealed that vascular SMCs respond to PDGF by synthesizing HA [88]. Together with its effect facilitating SMC migration as suggested by Waleh et al. and Gadelrb et al. [35,89], PDGF plays a supportive role in DA closure [90].

#### 2.2.4. The Role of IL-15 in Extracellular Matrix Production

Interleukin plays a role in diseases involving the dysregulation of ECM production, which is an essential factor of DA remodeling. Members of the interleukin family have been investigated for their role in DA closure. For example, elevated plasma levels of several interleukins, including IL-6, IL-8, IL-10, and IL-12, were associated with persistent PDA in premature infants [36]. In addition, emerging evidence demonstrates that IL-17 promotes vascular remodeling after atherosclerosis and affects prostaglandin expression, suggesting a potential effect in DA remodeling, which remains to be elucidated [91].

To date, IL-15 is the only interleukin proven to inhibit DA closure through the decrease in ECM production since IL-15 can diminish prostaglandin-induced HA production in the DA. Iwasaki et al. noticed genetic expressions of IL-15 were increased in DA compared with the aorta. They further demonstrated on rat DA that, although IL-15 itself cannot affect basal HA production, it can diminish prostaglandin-induced HA production dose-dependently [28].

These findings imply a possible relation between inflammation and hindered anatomical remodeling of the DA. Indeed, recent studies explored this correlation. Kim et al. discovered that intrauterine inflammation increased persistent PDA risk in extremely low birth weight infants treated with indomethacin [92]. Another meta-analysis also revealed a significant association between chorioamnionitis and PDA [93]. 

### 2.3. EC Proliferation

EC proliferation promotes anatomic closure via immunomodulation. Cytokines may contribute to such a mechanism, as tumor-derived cytokines promote the proliferation of vascular ECs [94]. Further study is warranted to clarify the role of cytokines in endothelial cell proliferation. Two correlating molecules are unraveled and discussed below. 

#### 2.3.1. Nitric Oxide

Nitric oxide (NO) is an agent that functions in various physiological processes, such as immune responses, inflammation, cell deaths, and vascular tone regulation [95]. NO is produced by the enzyme nitric oxide synthetase (NOS). Three types of NOS have been identified, including the endothelial NOS, the neuronal NOS, and the inducible NOS. Playing a role in functional closure, the genetic expression of NOS was found to be significantly altered in the DA during early development [61]. Endothelial NO, the most abundant type of NOS detected in the ductal wall, provided the strongest vasodilatory effect and was expressed not only in the luminal ECs but also in the vasa vasorum [96,97]. 

Aside from its role in functional closure, NO also inhibited EC proliferation during anatomical closure. Seidner et al. demonstrated through premature baboons that simultaneous inhibition of prostaglandin and NO stimulated anatomical remodeling of the DA [30]. The rate of EC proliferation was significantly higher when both prostaglandin and NO were inhibited, compared with that which only inhibited prostaglandin and that was not inhibited at all. In addition, Richard et al. discovered persistent eNOS expression in constricted mouse DA [98]. This suggests that NO also functioned during anatomical remodeling and played a role beyond vasodilation.

#### 2.3.2. The Role of Vascular Endothelial Growth Factor in Endothelial Cell Proliferation

Vascular endothelial growth factor (VEGF) is a signal protein secreted by cells promoting vasculogenesis and angiogenesis. Closely related to Notch, VEGF acts as a fundamental regulator for neovascularization [99]. VEGF’s appearance correlates with tissue hypoxia and contributes to vascular remodeling via the promotion of EC proliferation and migration. VEGF may induce inflammation and remodeling in bronchi cells, giving way to various lung diseases, such as pulmonary arterial hypertension, asthma, and even cancer [100,101].

VEGF promotes EC proliferation and assists in DA closure. Clyman et al. studied the effect of VEGF remodeling in premature baboons and sheep [31]. VEGF administration increased vasa vasorum ingrowth and thickened the neointima. Consistently, anti-VEGF antibodies were noted to inhibit vasa vasorum ingrowth and SMC migration during neointima formation. An inspiring study conducted by Sallmon et al. revealed the dissimilarity of VEGF and VEGF receptor expression induced by the two commonly used COX inhibitors, ibuprofen and indomethacin. Further investigation of such physiological differences is warranted [102].

#### 2.3.3. Angiotensin Type 2 

Angiotensin, a member of the renin–angiotensin system, is essential for normal renal development. For decades, the neonatal complications of pregnant mothers administrated with angiotensin-converting enzyme inhibitors or angiotensin receptor antagonists have been well established [103]. These complications include renal-related issues, such as oligohydramnios and renal failure, and disorders concerning other organ systems, such as respiratory distress syndrome and persistent PDA [104].

Angiotensin type 2 (AngII) correlates with a stimulatory effect on EC proliferation. Direct blockade of AngII receptor suppresses cell proliferation and restenosis post-angioplasty [105]. A proposed underlying mechanism is the interruption of neointimal proliferation. Bearing this in mind, Treszl et al. showed that the Ang II type 1 receptor’s specific genotype plays a role in PDA development [32]. The interference of neointimal proliferation may be the underlying mechanism for such PDA development.

### 2.4. IEL Disruption

The extent of internal elastic lamina disruption was noticed to have positive relations to SMC migration [106]. Furthermore, intimal thickening initiates at the site adjacent to IEL degeneration [75]. Similar IEL disruption mechanisms have been reported in systemic hypertension, atherosclerosis, and pulmonary hypertension [33]. Molecules influencing IEL disruption are described below.

#### 2.4.1. Chondroitin Sulfate and Dermatan Sulfate

Chondroitin sulfate (CS) and dermatan sulfate (DS) are carbohydrate–protein components of the arterial wall. CS has been discovered to participate in physiological phenomena such as cytokinesis and morphogenesis, and its association with pathological disorders and infection has been identified [107]. DS can assist heparin cofactor II in its inactivation of thrombin and reduce neointima formation after arterial injury [108]. Nevertheless, both CS and DS can form complexes with low-density lipoprotein and may play a role in atherosclerosis’s pathogenesis.

CS and DS facilitate IEL disruption in the DA. Hinek et al. displayed through fetal lamb DA SMCs that exaggerated levels of galactosamine-containing glycosaminoglycans, CS and DS, may cause shedding of the 67-kD receptor from the cell surface of SMCs, impairing the assembly of elastin in the DA, and ultimately leading to a complete disruption of the IEL [33]. The disruption of the IEL is associated with SMC migration into the subendothelial space, cellular proliferation, and alteration in collagen production, contributing to neointima formation [109].

#### 2.4.2. Tissue Plasminogen Activator

Tissue plasminogen activator (t-PA) is commonly recognized for its role in acute ischemic stroke. It is a protease converting plasminogen to plasmin. Plasmin, a protease, converts pro-matrix metalloproteinases (pro-MMPs) to MMPs, inducing tissue remodeling [110].

ECs can secrete t-PA to promote IEL disruption. Studying ECs isolated from the rat DA, Saito et al. demonstrated that t-PA secreted from DA ECs aids in the plasmin-induced activation of MMP-2 [34]. This activation is followed by IEL disruption, making up for intimal thickening in the DA. In fact, in an animal study, there was evidence showing higher levels of gene expression of t-PA in the DA compared with the aorta, in the term fetus or newborn rats [111]; however, no such evidence was observed in preterm infants.

Nonetheless, there is some evidence also revealing that t-PA contributed to the inhibition of hyperplasia and neointima formation. Wu et al. observed less intima thickness in injured arteries with significantly higher t-PA mRNA expression [112], implying that t-PA may have diverse roles in intimal thickening.

In addition to MMP-2, MMP-9 has been also implicated in the role of DA patency. Capoluongo et al. revealed an association between neutrophil gelatinase-associated lipocalin (NGAL), an enhancer of MMP-9 function, and persistent PDA [113]. They analyzed the epithelial lung fluid of 28 human neonates three days after birth and found a significant association between the levels of NGAL in the bronchoalveolar lavage fluid and persistent DA patency.

### 2.5. Blood Cell-Related Mechanisms

Other than structures of the DA itself, blood cells are also responsible for the DA’s closure. Vascular wall ischemia elicits an inflammatory response, activating mononuclear cells and promoting cell adhesion [96]. Cell adhesion molecules play a major role in these mechanisms. Such interaction can only occur when the luminal flow is nearly absent, which presents at later stages of DA closure [38].

#### 2.5.1. The Role of VEGF in Blood Cell-Related Mechanism

VEGF promotes monocyte adhesion in the DA. Monocytes display chemotaxis in response to VEGF, assisting DA closure [96]. Studies have revealed that VEGF is essential for monocytes to adhere to the ductus lumen. Decreased luminal flow enhances vascular cell adhesion molecule and VEGF expression, while VEGF further stimulates monocytes’ adherence to the endothelium [114]. Monocyte adhesion to the ductus lumen is crucial for VEGF-induced expansion of the neointima’s subendothelial layer [35]. When adhesion is blocked, expansion of the neointima cannot be activated even under higher VEGF levels. The suggests the major role of monocytes in response to local ischemic change.

#### 2.5.2. The Role of PDGF in Blood Cell-Related Mechanism

PDGF promotes platelet plug formation in the DA. PDGF is secreted primarily by mononuclear macrophages [115]. After vessel injury, PDGF induced vascular SMC migration into the neointima and their proliferation. Evidence has shown that PDGF-BB may contribute to injury-induced neointimal hyperplasia [116]. Furthermore, low PDGF levels were found to associate with persistent PDA [36]. A platelet plug formed after DA constriction assists closure of the residual lumen and promotes remodeling. Engur et al. revealed that infants with persistent PDA displayed lower PDGF levels after birth [37].

#### 2.5.3. Other Cytokines

Cytokines display diverse roles in DA closure. Cytokines promoting cell adhesion have been identified during the closure of the DA. Tumor necrosis factor-alpha (TNF-α), Interferon-γ, and CD154 have been noticed to enhance vascular cell adhesion protein-1 (VCAM-1) and E-selectin expression [38]. Interestingly, cytokines may contribute to PDA as well. In normal infants, postnatal DA constriction induces ischemia and increases cell death and ductus vasoreactivity loss [117]. However, in the condition of PDA, the DA remains responsive to vasoactive stimuli. Vasodilatory cytokines are independent of prostaglandin pathways, such as TNF-α and IL-6, to maintain the DA’s patency. The complex network of cytokines requires further investigation to clarify individual cytokines’ distinct features and their effect on DA patency modulation.

### 2.6. Cell Signaling Pathways

#### 2.6.1. Mitogen-Activated Protein Kinase (MAPK) Pathway

The MAPK family includes three major groups: ERK, p38, and JNK. In general, ERK1/2 is involved in cell growth, while p38 MAPK can regulate cell death, and JNK has been suggested to have not only essential roles in inflammation and apoptosis but also in cell migration and proliferation. Our previous study has demonstrated that in DA SMCs, Notch inhibition by DAPT can attenuate activations of ERK1/2 and JNK induced by Ang II [22], which reasoned its mechanisms underlying anti-proliferative and anti-migratory effects. In addition, we also found that BNP can exert anti-remodeling effects in the DA, with associated downregulation of ERK1/2 signaling [118].

The literature regarding the role of p38 in DA anatomical closure is limited. However, it has been shown that activation of the EGFR/p38/JNK pathway by mitochondrial-derived hydrogen peroxide contributes to oxygen-induced functional closure of the DA [119].

#### 2.6.2. Akt Pathway

Akt plays a role in the pathogenesis of vascular remodeling. Akt substrate GSK3β is a crucial protein in SMC proliferation where inhibition of GSK3β by Akt-induced phosphorylation increases SMC proliferation. It was shown that eNOS/Akt signaling can mediate vascular remodeling [120]. In line with this concept, Notch inhibition was found to inhibit DA SMC proliferation and migration with downregulation of Akt signaling [22].

## 3. Future Clinical Implications

Regulation of DA patency is not only controlled by vasoreactivity but also remod- eling. Thus, we suggest that molecules regulating remodeling pathways of the DA other than PG should be considered as potential pharmacologic targets in pre-clinical or clinical trials, such as Notch inhibitor, retinoic acid receptor antagonist, BNP, or NO activator. Furthermore, mixed mechanisms of pharmacological interventions might be a novel alternative strategy in clinical management, such as combined NO inhibitor and COX inhibitor indomethacin [30]. In addition, there might be different pathomechanisms between preterm and term infants. For example, the effect of platelets on ductal closure is minimal in term infants but it is more pronounced in preterm infants. Furthermore, vitamin A has also been shown to be effective in preterm but not full-term infants [121]. Thus, the approach of DA therapy necessitates individualized treatment.

## 4. Conclusions

Several molecules independent from prostaglandin pathways during anatomical closure have been recognized in this review. Five major groups of vascular remodeling mechanisms, including SMC migration/proliferation, ECM production, EC proliferation, IEL disruption, other blood cell-related mechanisms, and their correlated molecules, were discussed. These molecules may provide possible pharmacological targets in the future for either maintaining DA patency or inducing DA closure. Further study may broaden visions on the non-prostaglandin pathways and establish a more comprehensive and complete understanding of vascular remodeling orchestration.

## Figures and Tables

**Figure 1 ijms-22-03238-f001:**
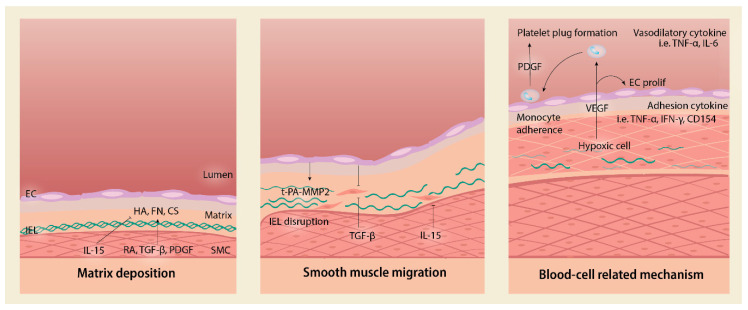
Sequential steps for intimal thickening. Ductal closure is initiated by matrix deposition by smooth muscle cells (SMCs) and endothelial cells (ECs). ECs secrete t-PA that activates MMP-2 and digest internal elastic laminae (IEL). SMC migration is mediated by numerous pathways that will be explored in Figure 2. The blood cell-related mechanism encompasses cells under hypoxic status to secrete VEGF and attract monocytes that mediate PDGF-mediated platelet plug formation. t-PA = tissue plasminogen activator; MMP = matrix metalloproteinase; FN = fibronectin; HA = hyaluronic acid; CS = chondroitin sulfate; RA = retinoic acid; TGF-β = transforming growth factor-beta; PDGF = platelet-derived growth factor; IL = interleukin; VEGF = vascular endothelial growth factor; EC = endothelial cell; IEL = internal elastic laminae; TNF = tumor necrosis factor; IFN = interferon; prolif = proliferation.

**Figure 2 ijms-22-03238-f002:**
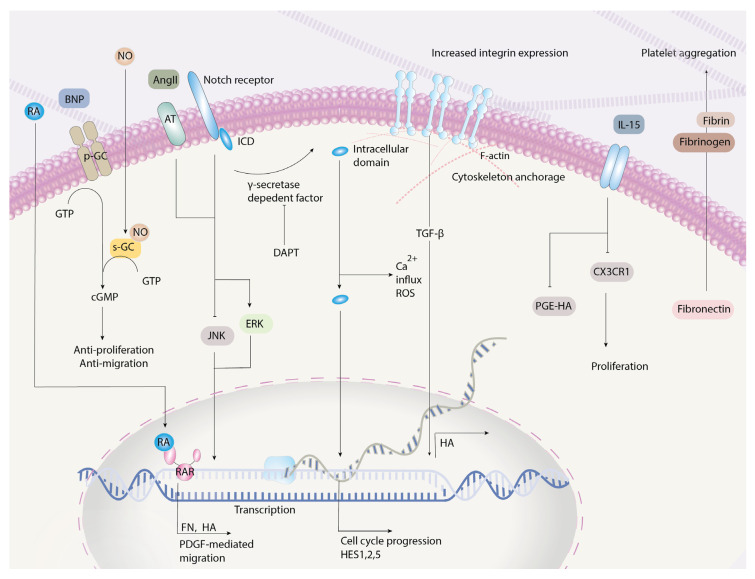
Molecular pathways of SMC-mediated migration and proliferation. The Notch system, upon secretase-dependent factor cleavage, releases the intracellular domains. The intracellular domains mediate ROS production, calcium influx, and ERK1/2 signaling, and act as transcription factors promoting cell cycle progression. TGF-β-mediated integrin expression increases anchorage to the SMC cytoskeleton and inhibits migration. Fibronectin is only effective when coupled with fibrin and fibrinogen, where the absence of fibrin and fibrinogen inhibits platelet aggregation and leads to a patent ductus. RA = retinoic acid; RAR = retinoic acid receptor; AngII = angiotensin II; AT = angiotensin receptor; MAPK = mitogen-activated protein kinases; ERK = extracellular regulated kinase; JNK = c-Jun N-terminal kinase; ICD = intracellular domain; PGE-HA = prostaglandin-induced hyaluronic acid production; PDGF = platelet-derived growth factor; FN = fibronectin; HA = hyaluronic acid; HES = hairy and enhancer of split; DAPT = N-[N-(3, 5-difluorophenacetyl-L-alanyl)]-S-phenylglycine t-butyl ester; TGFβ = transforming growth factor-beta; ROS = reactive oxygen species; IL = interleukin; NO = nitric oxide; BNP = B-type natriuretic peptide; pGC = particulate guanylyl cyclase; sGC = soluble guanylyl cyclase.

**Table 1 ijms-22-03238-t001:** Molecules involved in DA remodeling.

Mechanism	Molecule	Effect/Role	References
SMC Migration and Proliferation	Notch	Stimulates SMC migration and proliferation	[13,22]
	Fibronectin	Stimulates SMC migration	[23,24]
	Retinoic Acid	Stimulates SMC migration and proliferation	[25,26]
	TGF-β1	Inhibits SMC migration	[27]
	IL-15	Inhibits SMC proliferation	[28]
	PRDM6	Loss of function inhibits SMC proliferation	[29]
ECM Production	Retinoic Acid	Increases FN and HA production	[25]
	TGF-β	Increases HA and CS production	[21]
	PDGF-BB	Increases HA production	[21]
	IL-15	Decreases prostaglandin-induced HA production	[28]
	NO	Inhibits EC proliferation	[30]
EC Proliferation	VEGF	Promotes EC proliferation	[31]
	Angiotensin type 2	Promotes EC proliferation	[32]
IEL Disruption	CS and DS	Promotes IEL disruption	[33]
	t-PA	Promotes IEL disruption	[34]
Blood Cell-Related Mechanisms	VEGF	Promotes monocyte adhesion	[35]
	PDGF	Promotes platelet plug formation	[36,37]
	Cytokines	Promotes cell adhesion	[38]

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
