# Peer review of "Molecular Mechanisms Underlying Remodeling of Ductus Arteriosus: Looking beyond the Prostaglandin Pathway"

_ijms, 2021, doi:10.3390/ijms22063238_

Round 1
Reviewer 1 Report
The authors have organized information well that the diverse mechanism factors would possibly for DV closure. comments are below.
- Can you provide how associated hypertension and pulmonary hypertension in a direct mechanism of regional SMC of DV?
- tPA (2.4.2) has not associate directly with a premature infant.
Reviewer 2 Report
In this review paper, authors have highlighted the molecular mechanisms underlying remodelling of Ductus Arteriosus focusing on molecules independent from prostaglandin pathway. Th e review is very well written with good balance of information, tables, figure explaining the topic. This article falls under the scope of IJMS. I would like to recommend authors to address few issues before considering appropriate for publication
Comments
- Although authors have mentioned ERK and JNK in few place in the manuscript, they have missed another important MAPK family protein; p38.
It is known that activation of the EGFR/p38/JNK pathway by mitochondrial-derived hydrogen peroxide contributes to oxygen-induced contraction of ductus arteriosus
https://pubmed.ncbi.nlm.nih.gov/24906456/
Please discuss it.
- Authors have mentioned the role of MMPs, specially MMP-2 in remodelling/ECM degradation. There are 24 MMPs. How about other MMPs? Do they have role in Ductus arteriosus?
- line 181-182 “In contrast to the findings in the DA, atRA was found to suppress smooth muscle cell migration and proliferation in other vessels.” Please elaborate this. Does author mean atRA suppress SMC migration and proliferation in all other vessels except ductus arteriosus? If so, what property of atRA makes it so specific to inhibit SMC migration and proliferation in another vessel only (not DA)?
- Section 2.2.3 PDGF-BB. Please add study showing PDGF-BB promotes vascular smooth muscle cells migration and proliferation (example PMID 2701852). This reference ca b
- Section 2.2.4 IL-15 is too short (just a reference Iwasaki et al is not enough). Please discuss more paper to support its role in DA.
- Section 2.3 EC proliferation. This section is fine but how about the regulation of Ductus arteriosus by nitric oxide released by endothelial cells. Please discuss it. https://www.nature.com/articles/pr19982150
- I recommend author to add a new section or a new table to summarize “Cell signalling pathway (MAPK, eNOS/AKT, COX etc) involved in remodelling of DA”
- Texts are cryptic in many places. Needs extensive grammatical correction and spelling error correction.
Reviewer 3 Report
The authors present literature review of molecules involved in ductus arteriosus patency maintenance or closure beyond the prostaglandin pathway. The paper discusses the pathways and the corresponding mechanisms contributing to vascular remodeling. The manuscript focuses on search for molecules promising for future clinical implementation. Authors conclude that the identified molecules may be potential targets for future clinical management of ductus arteriosus patency. There are several suggestions to improve the quality of the manuscript.
- Authors should describe the molecular nature of Notch in more detail.
- Authors currently use the identical titles for subsections of the manuscript, for example, the titles for 2.1.2.1 and 2.2.4 subsections are the same: Interleukin-15. The same is true for the title of 2.1.1.3 and 2.2.1: Retinoic acid. There are some other almost identical titles. It is understandable that these subsections discuss different mechanisms, but having identical titles for different subsections is still confusing. Author should extend these titles with brief mentioning of the mechanism discussed. For example: “The role of interleukin-15 in ECM Production”, etc.
- The manuscript would benefit if authors identify the molecules (among the discussed ones) most promising for initiation of preclinical and clinical trials and speculate on possible outcomes in certain clinical situations. What would be the difference between the approaches to management of ductus arteriosus patency in babies born preterm and at normal gestational age?
- Authors should check style and grammar throughout the text. There are lots of spaces missing. The phrases such as “These development include the development of somite-derived organs...”, “Jagged1, a ligand the reacts with…”, and “...two stages of cleavage of occurs…” should be corrected/rephrased. There are typos in Table 1. Please check it.
Reviewer 4 Report
Summary of review:
The publication entitled “Molecular Mechanisms Underlying Remodeling of Ductus Arteriosus (DA): Looking Beyond the Prostaglandin Pathway » by Hsu HW et al. aimed at investigating in a review the molecules with a known vascular reactivity and their mechanisms independent from the prostaglandin pathway as potential targets for failure of DA patency.
The main strengths are the following:
The review is well in 5 chapters proposing the EC and SMC interactions in terms of both proliferation and migration. Two chapters introduce proteoglycan and T-PA and blood cells. Thus 17 mechanisms for a triad EC, SMC and blood cells have been reviewing and presented in a clear table 1.
The fig.1 introduces the normal ductal closure. And the figure is dedicated to an illustration of the chapter 1 without reporting PRDM6 mechanism.
The main weaknesses and/or lacks are the following:
The introduction is too short since the problem is failure to close the Patent DA. The vascular remodelling is essential but the normal functional closure is understood but the failure that occurs in 25 % of the preterm neonates is not discussed.
Cox inhibitors are essential are have been the subjects of RCT and meta-analysis but their activity could be limited by their known adverse side effects or also to the pathophysiological consequence or etiological causes of preterm situations. The introduction could introduce this balance of benefits and risks.
- Ouali in 2020 (frontiers in paediatrics) presented factors that maintain ductal patency in utero. This review should be mentioned as similar topics are introduced. Some factors like high levels adenosine, activation of K-channels cited by Ouali, but others sometimes outside gut microbiote, pluripotent stem cells, PCSK9 and lipoprotein metabolism. Biomarkers to guide their application of the molecular mechanisms proposed in the present review, for example in function of additional biomarkers with vitamin D, adenosine, serum lipids (LPS, TLR4 …) etc. Mitochondrial function is essential for the vascular reactivity is not presented here. Proteomic and gene approaches could complete the research on PDA.
For each chapter as the rule is a balance between risks and benefits, a conclusion exploring a strategy in function of the possible side effects should improve the reading. Many cardiovascular classical drugs have been investigated in the past but they failed to improve the PDA. Is there a priority of a mixed mechanism of pharmacological interventions ?
A short conclusion without mentioning the pharmacological targets that could be used in the future is not presented.
Minor points : All TGF- 1 in the table 1 are lacking their beta symbol of PDGF-BB is false.
In conclusion, this paper is interesting for its pertinence for human use but may be improved before publication.
Round 2
Reviewer 2 Report
Thank you addressing all comments. I have recommended for publications
Reviewer 3 Report
The corrections have been made by the Authors. The article may be published in the current form.
Reviewer 4 Report
The new manuscript has been reviewed and corrected now. All my points have been taken into consideration with appropriate answers and addition in the new manuscript.
Sorry for my mistake in reference Ouali/Ovali. The authors used the right one.
The review has been improved.